# Upregulation of Metrnl improves diabetic kidney disease by inhibiting the TGF-β1/Smads signaling pathway: A potential therapeutic target

Lu Lin[☯], Shulin Huang[☯], Xin Lin[☯], Xiaoling Liu, Xiangjin Xu, Chunmei Li, Pin Chen [ORCID] *

900TH Hospital of Joint Logistic Support Force (Fuzong Clinical Medical College of Fujian Medical University), Fuzhou, China

☯ These authors contributed equally to this work.
* chenpin@21cn.com

## Abstract

### Purpose

This study comprises an investigation of the role of meteorin-like (Metrnl) in an experimental model of diabetic kidney disease (DKD).

### Methods

Twenty-four db/db mice were randomly assigned to one of the following groups: DKD, DKD + Metrnl[-/-], and DKD + Metrnl[+/+]. Plasma Metrnl concentrations were measured using ELISA. Kidney tissues were examined via western blotting, qRT-PCR, and immunohistochemistry to determine the expression levels of inflammatory factors. Electron microscopy was employed to observe stained kidney sections.

### Results

Compared with the NC group, FBG, BW, and UACR were elevated in the DKD and Metrnl[-/-] groups, with severe renal pathological injury, decreased serum Metrnl concentration, decreased renal Metrnl expression, and increased expression levels of TNF-α, TGF-β1, TGF-R1, pSmad2, pSmad3, and α-SMA. In contrast, the Metrnl[+/+] group showed decreased FBG and UACR, BUN, TC and TG, increased HDL-C and serum Metrnl concentration, increased renal Metrnl expression, and decreased expression of TNF-α, TGF-β1, TGF-R1, pSmad2, pSmad3, and α-SMA, compared to the DKD and Metrnl[-/-] groups. A Pearson bivariate correlation analysis revealed a negative correlation between UACR and Metrnl, and a positive correlation between UACR and TGF-β1.

### Conclusion

Upregulation of renal Metrnl expression can improve renal injury by downregulating the expression of molecules in the TGF-β1/Smads signaling pathway in the renal tissues of

**Data Availability Statement:** All relevant data are within the article and its Supporting information files.

**Funding:** This study was supported by the National Natural Science Foundation of China (81870581) and Postdoctoral Special Fund of 900TH Hospital of Joint Logistic Support Force (49888).

**Competing interests:** The authors have declared that no competing interests exist.

**Abbreviations:** BUN, blood urea nitrogen; BW, body weight; cDNA, complementary DNA; DKD, diabetic kidney disease; ESKD, end-stage kidney disease; FBG, fasting blood glucose; HDL-C, high density lipoprotein cholesterol; Metrnl, meteorin-like; NC, normal control; NS, normal saline; OD, optical density; PAS, periodic acid-schiff; PASM, periodic acid-silver methenamine; qRT-PCR, quantitative real-time fluorescence PCR; TC, total cholesterol; TG, triglyceride; TGF-β, transforming growth factor-beta; TNF-α, tumor necrosis factor-alpha; UACR, urinary albumin-to-creatinine ratio.

type 2 diabetic mice; and by reducing the production of fibrotic molecules such as α-SMA.

---

## 1. Introduction

Diabetic kidney disease (DKD) is a chronic condition resulting from diabetes mellitus, representing a common microvascular complication of this disease [1]. DKD is a major contributor to end-stage kidney disease (ESKD) and has emerged as the leading cause of ESKD in both developed and developing nations [2, 3]. The pathogenesis of DKD is multifactorial, involving hemodynamic alterations, persistent low-grade inflammation, activation of the fibrotic factor cascade, and genetic as well as epigenetic abnormalities [4–6]. Although significant advancements have been made in elucidating the underlying mechanisms of DKD, effectively targeting these pathogenic pathways continues to pose a substantial challenge.

Recent studies have demonstrated a close relationship between irreversible kidney injury in type 2 diabetes and transforming growth factor-beta (TGF-β) [7]. Activation of the TGF-β1/Smads signaling pathway can lead to increased renal intrinsic extracellular matrix synthesis, decreased degradation, and a phenotypic shift of renal tubular mesangial fibroblasts towards myofibroblasts [8]. This ultimately results in diffuse glomerulosclerosis and interstitial fibrosis, significant aspects of renal fibrotic lesions in DKD [9]. Consequently, several research teams have explored interventions targeting this fibrotic pathway, achieving varying degrees of improvement in renal fibrosis with promising outcomes [10, 11]. Therefore, preventing excessive TGF-β1/Smads signaling pathway activation may represent a feasible and effective intervention against DKD. However, the key factors regulating the upstream mechanisms remain unclear.

Meteorin-like protein (Metrnl), a class of adipokines, exhibits various biological activities, and its active form can be induced by exercise or cold stimulation [12]. It is broadly distributed in the body and expressed to varying degrees across different tissues and organs. Metrnl modulates several biochemical processes, including neurodevelopment, white fat browning, and insulin sensitization [13, 14]. Studies have explored the relationship between Metrnl levels and adverse diabetic renal events in patients with DKD, revealing that serum Metrnl concentrations are inversely associated with the risk of DKD [14]. Additionally, persistently low levels of serum Metrnl have been shown to exacerbate adverse events such as glucose tolerance abnormalities and impaired endothelial function [15]. Despite these findings, the specific role of Metrnl in DKD remains insufficiently elucidated. Furthermore, there are indications that the in vivo expression level of Metrnl is associated with tumor necrosis factor-alpha (TNF-α) and inflammatory factors such as [12]. Both TNF-α and TGF-β are known to induce adverse events in diabetic kidneys, including inflammation and fibrosis [16, 17].

To address these knowledge gaps, we established a mouse model of DKD. This was achieved by administering a Metrnl-overexpressing adenovirus via the tail vein and subsequently observing changes in biochemical markers, renal injury indicators, and the expression of proteins associated with the TGF-β1/Smads pathway different groups of mice. Our main objective was to investigate the role of Metrnl in modulating the TGF-β1/Smads pathway in DKD and to assess its potential to improve renal injury in mice. The findings of this study may contribute to the identification of novel therapeutic targets for the treatment of DKD.

## 2. Materials and methods

### 2.1 Main materials and reagents

Cells: SV40 MES 13 mouse glomerular mesangial cells (Jaegi Biotech, ZCL1030); Mice: 8-week-old specific pathogen-free-grade db/db male mice and eight littermate control db/m mice (Jiangsu Jicui Pharmachem Bio-Tech Co., Ltd., China); Adenoviruses: Metrnl overexpression adenoviruses and Metrnl control adenoviruses (Fuzhou Jaegi Bio-Tech Co., Ltd., China), the adenoviral vectors designed to overexpress Metrnl in db/db mice systemically; Antibodies: TNF-α antibody (Proteintech, China); Metrnl antibody, TGF-β1 antibody, TGF-R1 antibody, p-Smad2 antibody, p-Smad3 antibody, α-SMA antibody, β-actin antibody (Abcam, UK). All animal experimentation processes strictly adhered to Regulations on the Management of Laboratory Animals. Animal experiments were approved by the Experimental Animal Welfare Ethics Committee of the 900th Hospital of the Joint Security Force (approval number: 2021–019). The report in this manuscript follows the recommendations of the ARRIVE guidelines. The euthanasia procedures were consistent with the AVMA guidelines.

### 2.2 Adenovirus titre cell validation assay

According to the MOI calculation formula for the template, which is defined as MOI (multiplicity of infection) = [viral titer (PFU/ml) × viral fluid volume]/ number of transfected cells, the optimal intervention load of the adenovirus was determined to be $2 \times 10^9$ PFU. This determination was based on previous literature [18, 19], pre-experiment results, and the safe range of adenovirus experiments, as provided by the Carrier Bioadenovirus Operation Manual. The specific groups were as follows: group A (blank control group) with an MOI of 0 (100ul PBS), group B with an MOI of 1000 (10ul viral stock solution + 90ul PBS), group C with an MOI of 2000 (20ul viral stock solution + 80ul PBS), and group D with an MOI of 5000 (50ul viral stock solution + 50ul PBS). The cells were transfected with adenovirus for 8–12 hours, and cell survival status and fluorescence expression were observed at 12, 24, and 48 h. The optimal MOI value was selected based on cell survival status. The selected MOI value was examined to achieve more than 80% infection of host cells at 48 h to ensure that the cells did not exhibit any significant toxic reactions.

### 2.3 Animal model construction and grouping

In this study, a type 2 diabetic mouse kidney injury model was established using 8-week-old male SPF-grade db/db mice and littermate control db/m mice. The mice were exposed to 12-hour light and dark cycles at an ambient temperature of 22˚C and maintained on a normal diet throughout the experiment. After a week of acclimatization, the mice were randomly assigned to different groups. Initially, 24 db/db mice were divided into DKD + normal saline (NS), DKD + control adenovirus (Metrnl$^{-/-}$), and DKD + Metrnl overexpression adenovirus (Metrnl$^{+/+}$) groups, each containing eight mice. Eight db/m mice were included in the normal control (NC) group.

Critical parameters such as fasting blood glucose (FBG), body weight (BW), urinary albumin-to-creatinine ratio (UACR), blood urea nitrogen (BUN), total cholesterol (TC), triglyceride (TG) and high density lipoprotein cholesterol (HDL-C) were assessed in the morning urine of each group of mice every four weeks. The criteria for successful development of diabetic nephropathy in the experimental group's mice were fasting blood glucose $\geq$ 11.1 mmol/L or random blood glucose $\geq$ 16.7 mmol/L for three consecutive days, along with a urinary albumin-to-creatinine ratio $\geq$ 30 μg/mg on over three occasions on non-consecutive days.

Among the twenty-four db/db mice in the DKD group, 20 successfully developed a DKD. Unfortunately, despite disinfectant and anti-infective treatments, some db/db mice died before adenoviral intervention due to prolonged high blood sugar levels and groin region ulcers. Specifically, there were six mice in the DKD + NS group, seven in the Metrnl$^{-/-}$ group, and seven in the Metrnl$^{+/+}$ group. During the adenoviral intervention phase, all db/db mice showed good survival without fatalities. Notably, no deaths were observed in the 8-mouse NC group throughout the experiment. All animal experimental procedures were conducted in accordance with the guidelines and approved by the Ethics Committee for Animal Experiments (Ethics No. 2021-019) of the Fuzhou General Clinical Medical College of Fujian Medical University.

## 2.4 Metrnl overexpression adenovirus and empty adenovirus tail vein injection intervention

Once modeling was completed, mice from both the DKD + Metrnl$^{-/-}$ and DKD + Metrnl$^{+/+}$ groups were administered adenovirus diluents with a loading capacity of $2 \times 10^9$ PFU via tail vein injection. Equal volumes of normal saline were given to the NC + NS and DKD + NS groups. The peak effect was generally observed on the third or fourth day after injection. As a result of the in vivo characteristics of adenovirus expression, the maximum effect was achieved three or four days after injection. The highest impact of the adenovirus intervention was observed on the seventh day after injection, and the in vivo adenovirus expression started to decline on the eighth day post-injection. Therefore, the intervention period for this study was set to one week.

The selected MOI value was examined to achieve more than 80% infection of host cells at 48 h to ensure that the cells did not exhibit any significant toxic reactions. The mice were anesthetized and euthanized on the eighth day after injection. The established criteria for implementing euthanasia in mice involve conditions observed at the end of the experiment or if the animals are nearing death before the experiment's conclusion. After reaching the end-point criteria, animals are allowed a period of approximately 10 minutes before euthanasia is performed. Upon the termination of the study, 20 mice were euthanized. In addition, 4 mice that exhibited chronic hyperglycemia and subsequent infections in the inguinal region, which led to death, were euthanized preemptively. Prior to the successful generation of a diabetic nephropathy mouse model, we measured the BW, FBG, UACR, BUN, TC, TG and HDL-C in the various groups biweekly. Post model induction, these parameters were monitored daily. For the induction of anesthesia, an intraperitoneal injection of 1% sodium pentobarbital was administered at a dosage of 100 mg/kg to ensure the mice were humanely euthanized. Subsequently, serum and kidney samples were collected from mice for further analysis (Fig 1).

## 2.5 General and biochemical indicator tests

Throughout the experiment, detailed records were kept on food intake, water consumption, urine output, and body weight of each group of mice. Mouse FBG levels were determined by measuring blood glucose levels in the tail vein using an ABBOTT glucose meter. The BW of the mice was closely monitored using an electronic analytical balance. Microalbuminuria and urinary creatinine levels in mice were analyzed using the Roche Cobas 8000 c702 fully automated biochemical immunoanalyzer. Urinary albumin was standardized to urinary creatinine, specifically the UACR in μg/mg creatinine. The BUN, TC, TG and HDL-C were measured by AU2700 automatic biochemistry (colourimetric method).

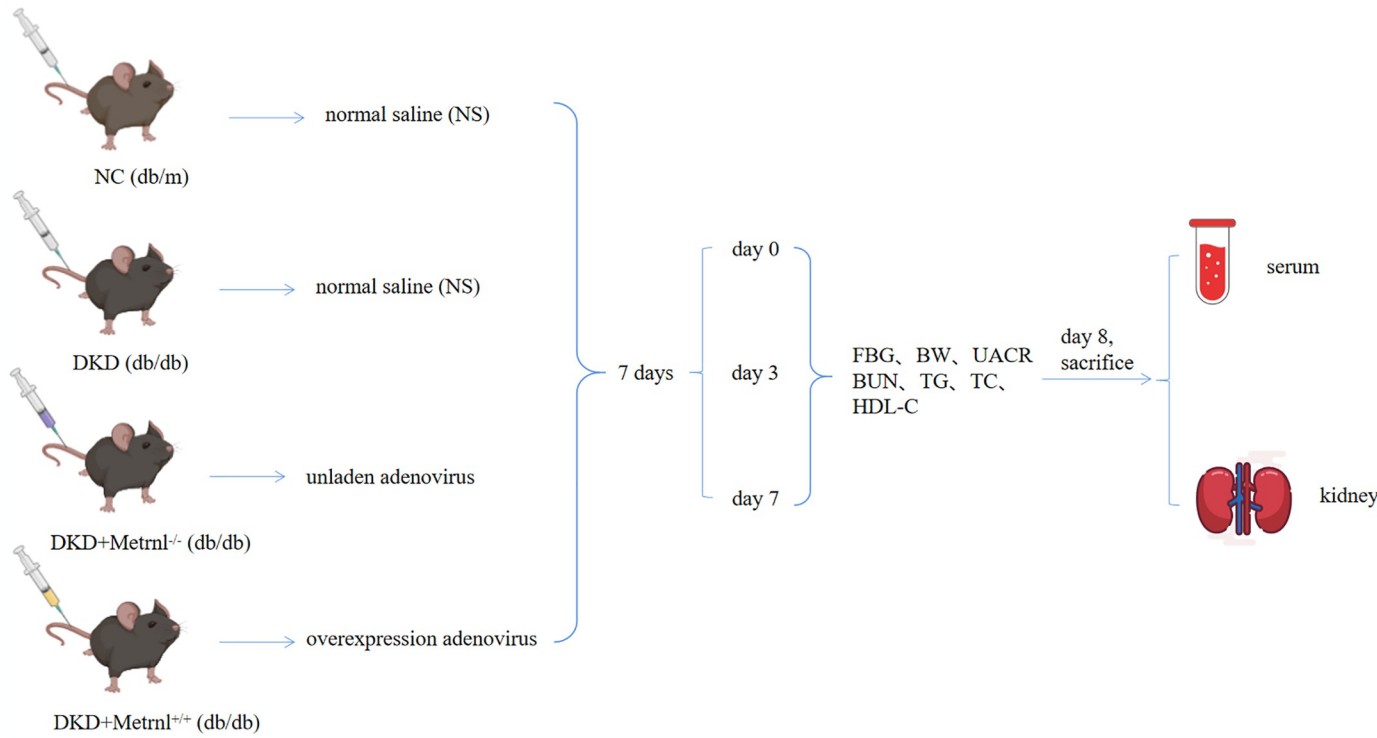

**Fig 1. The procedure of mouse modeling test.**

## 2.6 Enzyme-linked Immunosorbent Assay (ELISA)

The procoagulant tubes were maintained at room temperature for 20 min. Subsequently, the serum was separated by centrifugation for five minutes and placed in EP tubes. The standards were prepared according to the guidelines outlined in the instruction manual for the Mouse Meteorin-like/METRNL PicoKine ELISA Kit provided by Wuhan PhD. Upon completion of the TMB reaction, 100 μl of the reaction termination solution was carefully added to each well to halt the reaction without exposing it to light. The optical density (OD) of each well was measured at 450 nm using an enzyme marker.

The OD value of each well was measured at 450 nm using an enzyme marker. An enzyme marker was used to obtain absorbance values. The absorbance of the control well (0 pg/ml) was determined from the absorbance of each sample well. The vertical axis represents the absorbance value, whereas the horizontal axis represents the concentration. A standard curve was plotted to determine the concentration corresponding to the detection factor. The actual concentration, which was diluted N times, was calculated by multiplying the measured concentration with N.

## 2.7 Protein immunoblotting analysis (Western blot)

Approximately 50 mg of kidney tissue from the mice was dissected and lysed, followed by centrifugation. The protein supernatant was collected, and the BCA working solution was prepared according to the Pierce™ BCA Protein Assay Kit instructions. The concentration of each protein sample was determined by measuring the absorbance (OD value) and comparing it to a standard curve, with the concentration calculated after dilution. Protein spiking, electrophoresis, electrotransformation, and containment were performed. The primary antibody was

incubated overnight at 4°C before incubation with the secondary antibody the following day. An ultrasensitive ECL luminescent liquid was prepared and the target protein bands were analyzed by exposure using a fully automated chemiluminescence image analyzer.

## 2.8 Quantitative Real-time fluorescence PCR (qRT-PCR)

Approximately 50 mg of kidney tissue extracted from a mouse was placed into RNAse-free EP tubes containing Trizol solution for grinding. The RNA was subsequently purified using an all-in-one gold RNA extraction kit. RNA concentration was determined using a spectrophotometer. Based on RNA concentration, a transcription system was prepared, and RNA was reverse-transcribed into complementary DNA (cDNA). The cDNA was then used for quantitative real-time polymerase chain reaction (qRT-PCR) reactions, following the primer sequences of the relevant genes (Table 1). Finally, β-actin was used as an internal reference, and the $2^{-\Delta\Delta}Ct$ method was employed to calculate the relative levels of target gene expression.

## 2.9 Immunohistochemical (IHC) staining

Kidney tissues were sliced, dewaxed, hydrated, and baked. Goat serum was added dropwise to block non-specific binding sites, followed by incubation at room temperature. The primary antibody was added dropwise and the tissues were incubated overnight at 4°C in a refrigerator. The following day, a secondary antibody reaction enhancer was added dropwise, followed by incubation and the addition of enzyme-labelled goat anti-rabbit IgG polymers. The samples were incubated at room temperature. Subsequently, DAB color-developing solution was added dropwise, and the sample was re-stained with hematoxylin violet solution. The tissues were returned to PBS solution to remove any excess blue color and then rinsed. The dehydration and transparency steps were performed, followed by tissue coverage. Finally, the expression of Metrnl, TNF-α, TGF-β, and α-SMA proteins in mouse kidney tissues was observed and photographed under a microscope. Image-pro Plus version 6.0 professional Image analysis software (Media Cybernetics Company, Silver Spring, MD) was used to analyze the images of immunohistochemical staining sections.

## 2.10 Hematoxylin-eosin histochemical staining (HE)

Kidney tissue samples were cut, cleared of wax, hydrated, and fixed by baking. Subsequently, hematoxylin and eosin staining was conducted, and the tissues were dehydrated, clarified, and blocked before microscopic observation. Image-Pro Plus 6.0 professional image analysis

**Table 1. Summary of the primers used to measure mRNA expression.**

| Gene | Sequence |
| --- | --- |
| Mouse-Metrnl forward | ACCAGTGACTTTGTTGTCCGA |
| Mouse-Metrnl reverse | CACCCGCAGGTAGATGACTG |
| Mouse- TNF-α forward | GATCGGTCCCCAAAGGGATG |
| Mouse- TNF-α reverse | CCACTTGGTGGTTTGTGAGTG |
| Mouse- TGF-β1 forward | ACGTGGAAATCAACGGGATCA |
| Mouse- TGF-α1 reverse | GTTGGTATCCAGGGCTCTCC |
| Mouse-α-SMA forward | GCATCCACGAAACCACCTATAAC |
| Mouse-α-SMA reverse | ACAGAGTACTTGCGTTCTGGAG |
| Mouse-β-actin forward | CACTGTCGAGTCGCGTCCA |
| Mouse-β-actin reverse | CATCCATGGCGAACTGGTGG |

software was applied to analyse the images pathological sections (at ×400 magnification). In each stained section, 10 glomeruli were randomly selected, and the glomerular area and the area of positive staining in the glomeruli were determined. The mesangial matrix index, i.e., the area of positive staining/glomerular area, was calculated, and then the mean of the mesangial matrix index of 10 glomeruli was calculated, which was the relative amount of membrane matrix in the glomeruli of each section. The number of renal tubular infiltrating inflammatory cells was counted through 5 fields of view (×100) of the microscope.

## 2.11 Masson staining for collagen fibres

Kidney tissues were sliced, dewaxed, hydrated, and baked. The sections were immersed overnight in Masson A solution using a Masson three-color staining solution kit. They were washed the next day, and a mixture of Masson B solution and Masson C solution was prepared in equal proportions. The sections were immersed in the mixture, rinsed in water, and differentiated using 1% hydrochloric acid and alcohol. They were then immersed in Masson's D solution, rinsed under running water, immersed in Masson's E solution, and rinsed in 1% glacial acetic acid after staining. The sections were stained with Masson's F solution and rinsed with 1% glacial acetic acid for differentiation. Finally, the sections were dehydrated, made transparent, covered, and observed under a microscope to analyze collagen fibers in the renal tissues.

## 2.12 Periodic Acid-Schiff (PAS) staining

Kidney tissues were hydrated and dewaxed. The sections were then stained using the PAS Staining Solution Kit. They were first immersed in PAS Staining Solution B for 10 min, followed by water washing, before being immersed in PAS Staining Solution A avoid light exposure. The sections were then washed with water before being immersed in PAS Staining Solution C. After washing with water, the sections were differentiated using an aqueous solution of hydrochloric acid. After washing the sections with PBS to restore the blue color, the proliferation of basement membranes in mouse kidney tissues was observed under a microscope.

## 2.13 Periodic Acid-silver Methenamine (PASM)

The kidney tissue sections were dewaxed and hydrated. The tissue sections were stained using the PASM Staining Solution Kit. This was followed by microscopic observation of proliferation of the basement membrane and mesangial stroma in mouse kidney tissue.

## 2.14 Histopathological electron microscopic filming and electron microscopic observation

Renal cortex samples (measuring 1 mm × 1 mm×1 mm) were fixed by immersion in an electron microscope fixative and maintained at 4˚C for fixation. Afterwards, the samples were rinsed with 0.1M phosphate buffer with a pH of 7.4, followed by fixation of the renal tissue. The kidney tissue was rinses in 0.1M phosphate buffer with a pH of 7.4, dehydrated, infiltrated, embedded, and sectioned. After staining, the tissue sections were left to air dry overnight at room temperature. Subsequently, we used the transmission electron microscope to examine the ultrastructural features of the glomerular basement membrane, peduncle, nucleus, and mitochondria in each set of kidney tissue samples.

## 2.15 Statistical analyses

Data analysis was performed using the GraphPad Prism 8. Measurements are presented as means ± standard deviation (mean ± SD). One-way ANOVA was used to compare multiple groups, whereas two-way comparisons were assessed using the Tukey Honestly Significant Difference method. Paired t-tests were used for pre- and post-intervention comparisons. Furthermore, our analysis employed the Spearman correlation coefficient to conduct correlation analyses based on the normal distribution of the bivariate data. A statistical significance threshold of $P < 0.05$ was applied in our results.

# 3. Results

## 3.1 General and biochemical findings

Mouse glomerular mesangial cells transfected with adenovirus presented varying levels of green fluorescence expression in different groups: Group A (MOI: 0, corresponding to a viral load of $10 \times 10^9$ PFU), Group B (MOI: 1000, corresponding to a viral load of $5 \times 10^9$ PFU), Group C (MOI: 2000, corresponding to a viral load of 210^9 PFU), and Group D (MOI: 5000, corresponding to a viral load of 0 PFU). Notably, a clear difference in green fluorescence expression was observed between Group C (MOI: 2000, equivalent to $2 \times 10^9$ PFU) and Group D (MOI: 5000, equivalent to 0 PFU). The most robust fluorescence was observed in Group C, indicating the most effective transfection outcome and optimal cell viability (Fig 2A). As a result, a viral load of $2 \times 10^9$ PFU was chosen as the transfected virus's viral load for this study.

Before adenoviral intervention, the BW and FPG levels of mice in the DKD + NS, Metrnl$^{-/-}$, and Metrnl$^{+/+}$ groups consistently exhibited significant elevations compared to those in the NC + NS group ($P < 0.05$) (Fig 2B and 2C). On the normal diet, FBG levels steadily increased in all three groups of db/db mice and plateaued at 12 weeks of age. After treatment, the mice in each group displayed the distinct characteristics described above. However, the Metrnl$^{+/+}$ group demonstrated a significant reduction in FBG levels compared with the DKD + NS and Metrnl$^{-/-}$ mice ($P < 0.05$).

Similarly, the morning UACR of mice in the DKD + NS, Metrnl$^{-/-}$, and Metrnl$^{+/+}$ groups was higher than that of the NC + NS group before treatment ($P < 0.05$). Over time, the morning urine UACR of db/db mice in all three groups gradually increased, with microalbuminuria appearing at 12 weeks of age and substantial albuminuria manifesting at 20 weeks of age. After the intervention, the UACR, BUN, TC and TG in the Metrnl$^{+/+}$ group were significantly lower than that in the DKD and Metrnl$^{-/-}$ mice ($P < 0.05$), but remained higher than that in the NC + NS group ($P < 0.05$) (Fig 2D–2G). In contrast, mice in the DKD + NS, Metrnl$^{-/-}$, and Metrnl$^{+/+}$ groups had lower HDL-C than those in the NC + NS group before treatment ($P < 0.05$). After intervention, HDL-C was lower in the Metrnl$^{+/+}$ group than in the DKD and Metrnl$^{-/-}$ mice ($P < 0.05$) (Fig 2H).

## 3.2 Metrnl expression and effects in DKD mice

Metrnl expression was significantly reduced in kidney tissues of DKD mice. Western blotting and qRT-PCR analyses revealed a decrease in Metrnl protein and mRNA levels in the kidney tissues of the DKD, Metrnl$^{-/-}$, and Metrnl$^{+/+}$ groups compared to the NC + NS group. Metrnl expression was significantly higher in the Metrnl$^{+/+}$ group than in both the DKD and Metrnl$^{-/-}$ groups ($P < 0.05$) (Fig 3A, 3B and 3D).

The ELISA results were consistent with the reduced Metrnl expression observed in kidney tissues, as serum levels of Metrnl were significantly lower in both the DKD and Metrnl$^{-/-}$ groups than in the NC + NS group ($P < 0.05$). In contrast, the Metrnl$^{+/+}$ group had higher

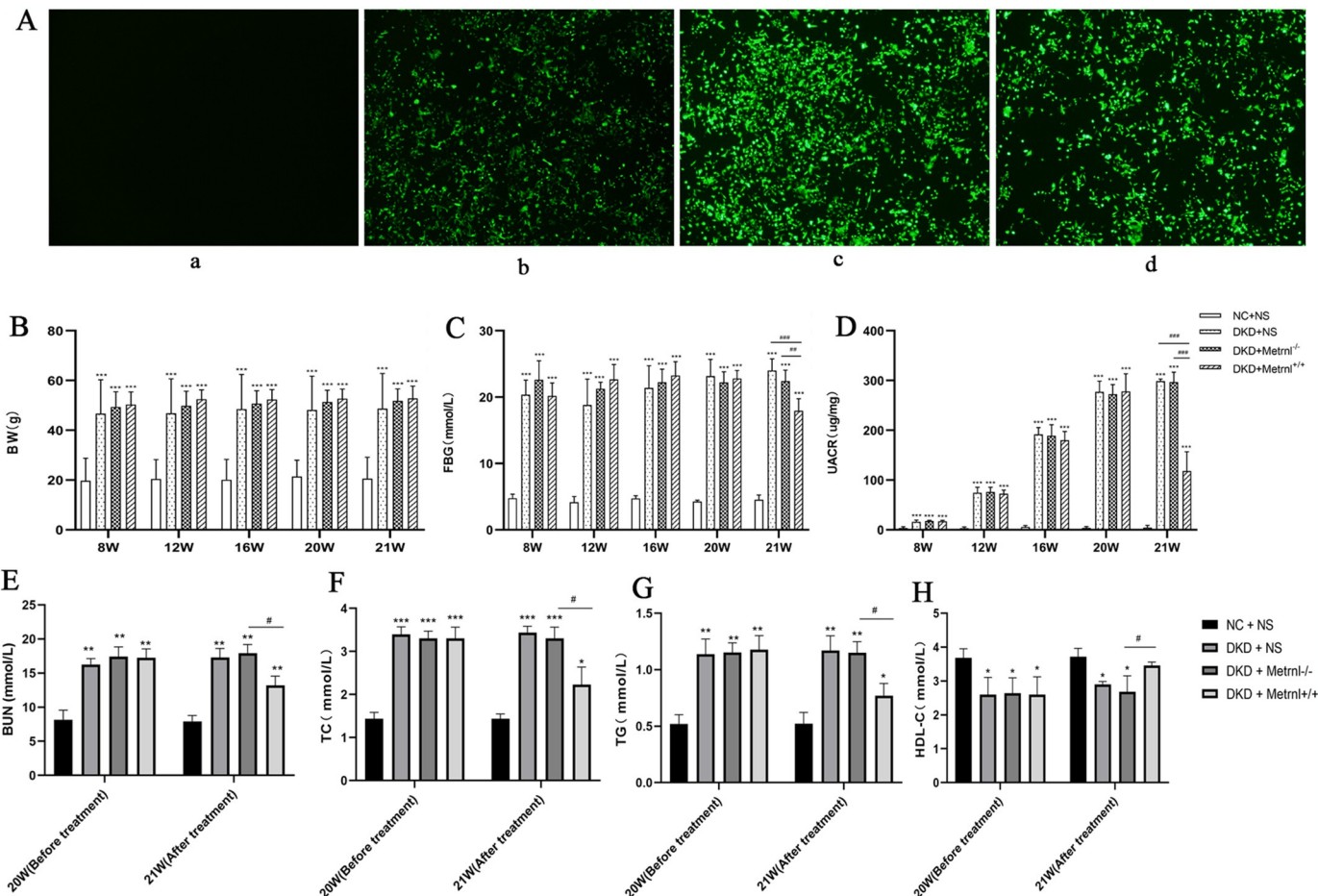

**Fig 2. Fluorescence expression in glomerular mesangial cells following adenoviral transfection and the assessment of basic indices in each group of mice.** (A) The fluorescence expression of mesangial adenovirus was evaluated in each group after transfection (a: MOI = 0, b: MOI = 1000, c: MOI = 2000, d: MOI = 5000), Scale bar = 50μm; Comparisons were made among groups of mice for body weight (B), fasting blood glucose (C), UACR (D), BUN (E), TC (F), TG (G) and HDL-C (H). NC + NS group: normal control group, DKD + NS group: diabetic nephropathy + saline group, DKD + Metrnl$^{-/-}$ group: diabetic nephropathy + control adenovirus group, DKD + Metrnl$^{+/+}$ group: diabetic nephropathy + Metrnl overexpression adenovirus group. Compared with the NC group, $^*P<0.05$, $^{**}P<0.01$, $^{***}P<0.001$; Compared with DKD + Metrnl$^{+/+}$ group, $^#P<0.05$, $^{##}P<0.01$, $^{###}P<0.001$, $^{####}P<0.0001$.

serum Metrnl levels than the NC + NS group (P < 0.05). Notably, no significant difference in the serum levels of Metrnl was observed between the DKD and Metrnl$^{-/-}$ groups (P > 0.05). Furthermore, both groups exhibited lower levels of Metrnl than the Metrnl$^{+/+}$ group (P < 0.05) (Fig 3C).

Immunohistochemical analysis of mouse kidney tissues revealed reduced positive staining of glomerular Metrnl protein in the DKD, Metrnl$^{-/-}$, and Metrnl$^{+/+}$ groups compared to that in the NC + NS group (P < 0.05). Interestingly, the group with Metrnl$^{+/+}$ exhibited a larger positive staining region for glomerular Metrnl than the DKD and Metrnl$^{-/-}$ groups (P < 0.05) (Fig 3E and 3F).

### 3.3 Up-regulation of Metrnl expression levels improves histopathology and ultrastructure of the kidney in DKD mice

As illustrated in Fig 4A and 4B, HE staining data showed enlarged glomerular morphology in the DKD and Metrnl$^{-/-}$ groups, which considerably increased the number of mesangial cells

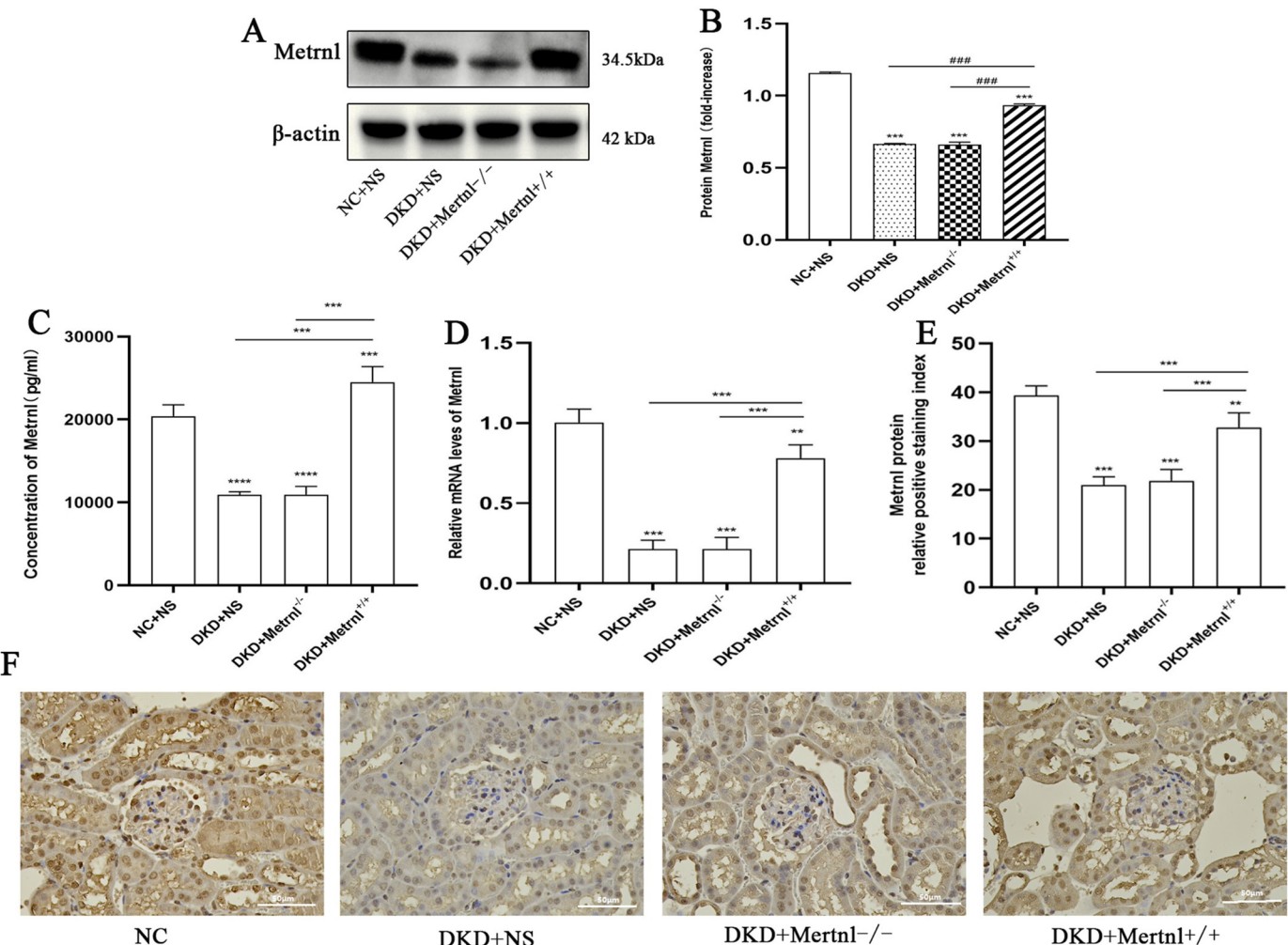

**Fig 3. Expression and role of Metrnl in DKD mice.** (A, B) Western blot analysis was performed to detect the expression levels of the Metrnl protein in mouse kidney tissues from each group. (C) ELISA was performed to measure the serum concentration of Metrnl in each group of mice. (D) Relative expression of Metrnl in the kidney tissues of each group was determined. (E, F) Immunohistochemical analysis was performed to detect glomerular Metrnl protein expression in each group of mice. Scale bar = 50μm. NC + NS group: normal control group, DKD + NS group: diabetic nephropathy + saline group, DKD + Metrnl$^{-/-}$ group: diabetic nephropathy + control adenovirus group, DKD + Metrnl$^{+/+}$ group: diabetic nephropathy + Metrnl overexpression adenovirus group. Compared with the NC group, *P<0.05, ** P<0.01, ***P<0.001; Compared with DKD + Metrnl$^{+/+}$ group, #P<0.05, ##P<0.01, ###P<0.001, ####P<0.0001.

(black arrow). In addition, the tubulointerstitial spaces of the renal tubules showed inflammatory cell infiltration (green arrow). Nevertheless, the glomerular morphology of the Metrnl$^{+/+}$ mice did not differ significantly from that of the NC + NS group. Additionally, there was no apparent inflammatory infiltration of the mesangial cells or interstitial spaces of the renal tubules. Compared with the DKD and Metrnl$^{-/-}$ groups, the glomerular morphology of mice in the Metrnl$^{+/+}$ group decreased, mesangial cell proliferation decreased (black arrow), and the number of renal interstitial inflammatory cells decreased (green arrow). The degree of proliferation of glomerular mesangial stroma was represented by the mesangial matrix index. Compared with the NC group, the mesangial matrix index and the number of renal tubular infiltrating inflammatory cells were significantly higher in the DKD group (Fig 4E and 4F, P < 0.05); And the mesangial matrix index and the number of renal tubular infiltrating

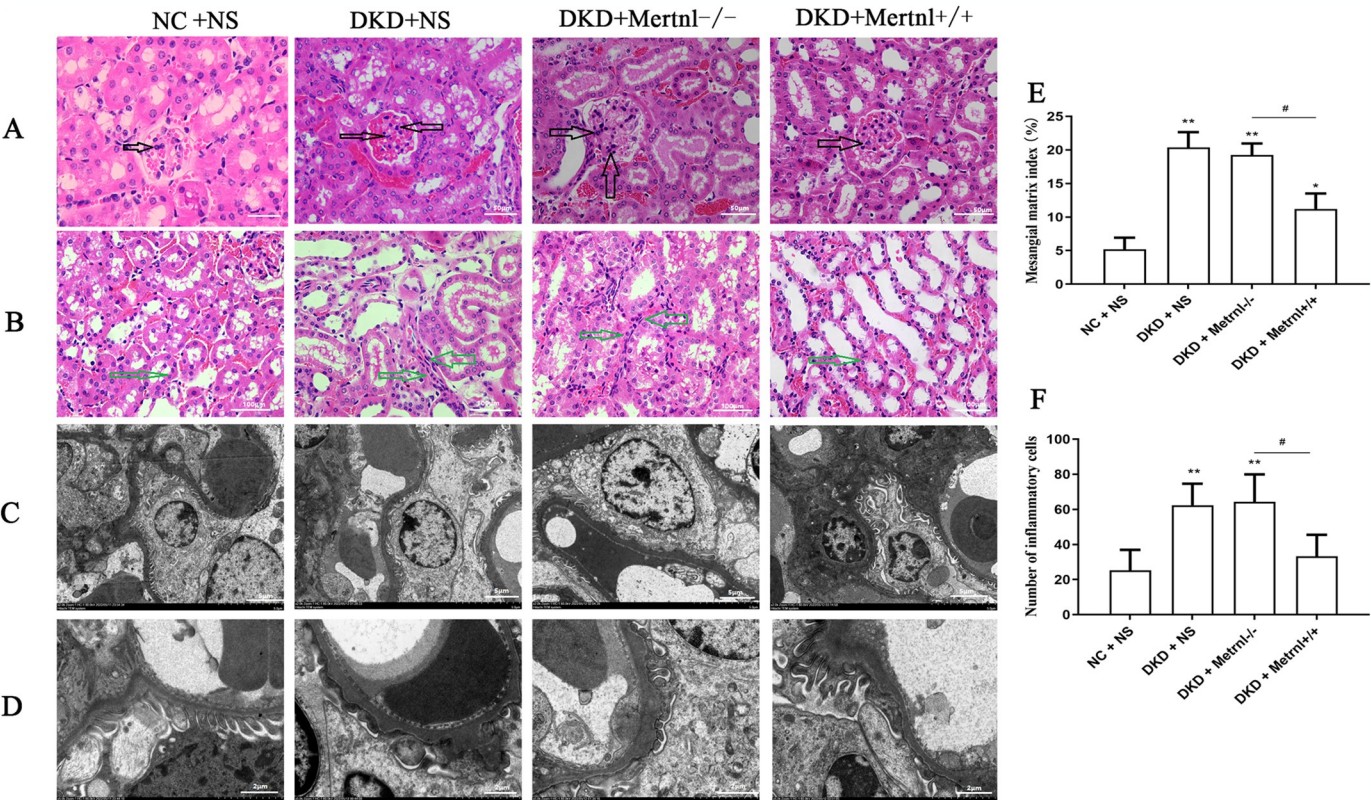

**Fig 4. Pathological changes in the kidneys of mice in various groups were assessed using HE staining and transmission electron microscopy.** (A) Glomerular HE staining. Scale bar = 50μm. (B) HE staining of renal tubules. Scale bar = 100μm. Transmission electron microscopy was used to examine the pathological changes in the kidneys of various groups at magnifications of 5μm (C) and 2μm (D). The mesangial matrix index (E) and the number of renal tubular infiltrating inflammatory cells (F). The Mesangial cells (black arrow) and inflammatory cells infiltration (green arrow). NC + NS group: normal control group, DKD + NS group: diabetic nephropathy + saline group, DKD + Metrnl$^{-/-}$ group: diabetic nephropathy + control adenovirus group, DKD + Metrnl$^{+/+}$ group: diabetic nephropathy + Metrnl overexpression adenovirus group.

inflammatory cells were significantly lower in the Metrnl$^{+/+}$ group compared with the Metrnl$^{-/-}$ group (Fig 4E and 4F, P < 0.05).

As depicted in Fig 4C and 4D, transmission electron microscopy of mouse kidney tissues showed that the basement membrane of the NC + NS group's mouse kidney tissues had a consistent thickness and remained undamaged and uninterrupted, with no noticeable ruptures or thickening. The majority of peduncles exhibited even thickness and a plethora of numbers. The nuclei of the peduncle cells featured a regular morphology with intact nuclear membranes, whereas the cells contained mitochondria with cristae. In contrast, the basement membrane of the kidney tissues of DKD and Metrnl$^{-/-}$ mice showed uneven thickness. The peduncles were fused and widened, with some swelling, and the stroma appeared shallow. Additionally, the nuclei of the peduncle cells were irregularly shaped and locally depressed.

Moreover, there was a shortage or absence of intracellular mitochondria-containing cristae. However, compared to the DKD and Metrnl$^{-/-}$ groups, the kidney tissue basement membrane in the Metrnl$^{+/+}$ group demonstrated thinning and increased peduncle count, while morphology tended towards normalcy. Furthermore, mitochondria-containing cristae were observed in peduncle cells. Nevertheless, the peduncle cell nuclei persisted, displaying irregularities and sinking indications.

### 3.4 Up-regulation of Metrnl expression level ameliorates glomerular collagen fibre proliferation and basement membrane alterations in DKD mice

Masson assay results for mouse kidney tissues indicated glomerular collagen fiber hyperplasia in the DKD and Metrnl$^{-/-}$ groups compared to the NC + NS group. However, no hyperplasia was observed in the Metrnl$^{+/+}$ group, and there was no significant difference compared with the NC + NS group. In addition, the Metrnl$^{+/+}$ group showed a reduction in glomerular collagen deposition compared with the DKD and Metrnl$^{-/-}$ groups (Fig 5A).

Similarly, the PASM findings in mouse kidney tissues indicated that the glomerular basement membranes of mice in the DKD and Metrnl$^{-/-}$ groups were thicker than those in the NC + NS group. Moreover, both mesangial cells and stroma showed proliferation. In contrast, the glomerular basement membranes of mice in the Metrnl$^{+/+}$ group displayed only slight

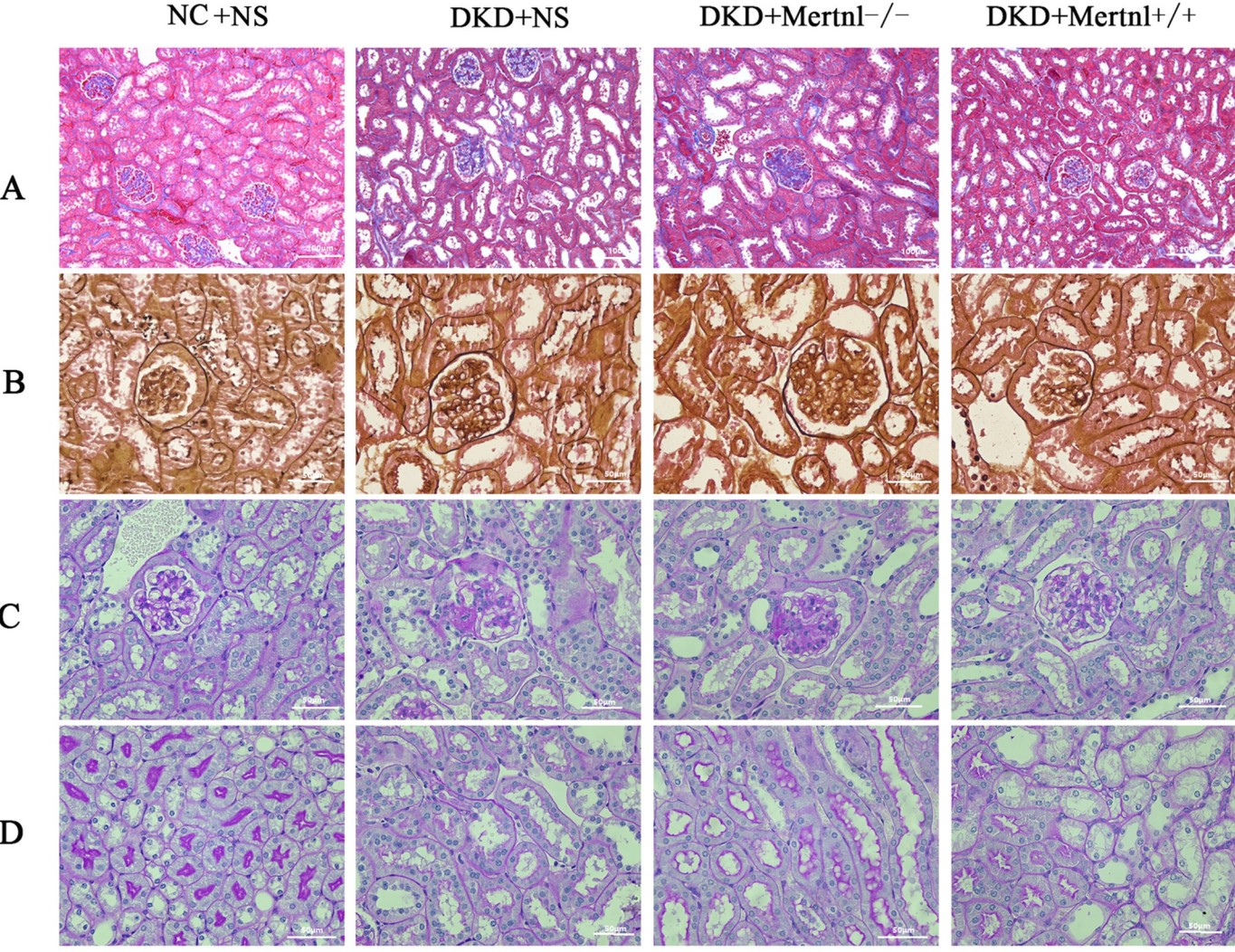

**Fig 5. Light microscopy examination was conducted to evaluate pathological changes in the kidneys of mice in each group using Masson staining, PASM staining, and PAS staining.** (A) Masson staining, scale bar = 100μm. (B) PASM staining; scale bar = 50μm. (C, D) PAS staining; scale bar = 50μm. NC + NS group: normal control group, DKD + NS group: diabetic nephropathy + saline group, DKD + Metrnl$^{-/-}$ group: diabetic nephropathy + control adenovirus group, DKD + Metrnl$^{+/+}$ group: diabetic nephropathy + Metrnl overexpression adenovirus group.

thickening, without any observed proliferation of mesangial cells and stroma. Additionally, compared to the DKD and Metrnl$^{-/-}$ groups, the thickness of the glomerular basement membrane was reduced in the Metrnl$^{+/+}$ group, with improved proliferation of mesangial cells and stroma (Fig 5B).

The PAS results of mouse kidney tissues revealed that compared to mice in the NC + NS group, the glomerular basement membranes in the DKD and Metrnl$^{-/-}$ groups were thickened. Additionally, there was an increase in the number of mesangial cells and stroma, leading to the formation of K-W nodules. Furthermore, the basement membranes of the renal tubules were thicker in these groups. In contrast, the basement membranes of the glomeruli and tubules in the Metrnl$^{+/+}$ group showed only slight thickening, and there was no evident formation of K-W nodules in the mesangial cells or stroma. The basement membrane of the glomeruli and tubules in the Metrnl$^{+/+}$ group exhibited slight thickening, and there was no proliferation of mesangial cells and stroma or any noticeable K-W nodule formation. Compared to the DKD and Metrnl$^{-/-}$ groups, the thickness of the glomerular and tubular basement membranes was reduced in the Metrnl$^{+/+}$ group, and there was a decrease in the degree of mesangial cell and stroma proliferation (Fig 5C and 5D).

### 3.5 Metrnl nephroprotective effect is associated with down-regulation of TGF-β1/ Smads signaling pathway and inhibition of α-SMA fibrosis molecule expression in renal tissues of DKD mice

The Western blot and qRT-PCR findings for mouse kidney tissues indicated that the expression of TNF-α, TGF-β1, TGF-R1, pSmad2, pSmad3, and α-SMA proteins was increased in the DKD, Metrnl$^{-/-}$, and Metrnl$^{+/+}$ groups compared to the NC + NS group (P < 0.05). Moreover, the relative mRNA expression levels of these factors were also increased (P < 0.05). In contrast, the kidney tissues of mice in the Metrnl$^{+/+}$ group exhibited elevated levels of TNF-α, TGF-β1, TGF-R1, pSmad2, pSmad3, and α-SMA protein and mRNA expression levels compared to the DKD and Metrnl$^{-/-}$ groups. However, these levels decreased when compared to the DKD and Metrnl$^{-/-}$ groups, with a significance of P < 0.05 (Fig 6A–6E).

Immunohistochemical detection in mouse kidney tissues showed a significant increase in positive staining for TNF-α, TGF-β1, and α-SMA proteins in the glomeruli of mice in the DKD, Metrnl$^{-/-}$, and Metrnl$^{+/+}$ groups compared to that in the NC + NS group (P < 0.05). The Metrnl$^{+/+}$ group displayed a reduced range of positive staining for glomerular TNF-α, TGF-β1, and α-SMA proteins compared to the DKD and Metrnl$^{-/-}$ groups (P < 0.05) (Fig 6F). As shown in Fig 6G, Pearson's bivariate correlation analysis revealed that the urine albumin-to-creatinine ratio (UACR) was significantly negatively correlated with the percentage of the Metrnl-positive area in immunohistochemical renal tissues (P < 0.05). Additionally, UACR positively correlated with the percentage of TGF-β1-positive cells (P < 0.05). The correlation coefficients for these correlations were -0.9493 and 0.9428, respectively. Furthermore, the percentage of Metrnl-positive area was significantly negatively correlated with the percentage of TGF-β1 positive area, with a correlation coefficient of -0.9144 (P < 0.05).

## 4. Discussion

In this study, we developed a DKD model in type 2 diabetic mice and observed a significant decrease in serum Metrnl expression in DKD mice compared to normal mice. We investigated the effects of Metrnl overexpression on the progression of DKD and renal pathology. Our results indicated that overexpression of Metrnl led to a decreased UACR and ameliorated renal pathological damage in DKD mice.Additionally, we examined the TGF-β1/Smads signaling pathway in the context of renal fibrosis associated with DKD. We found substantial

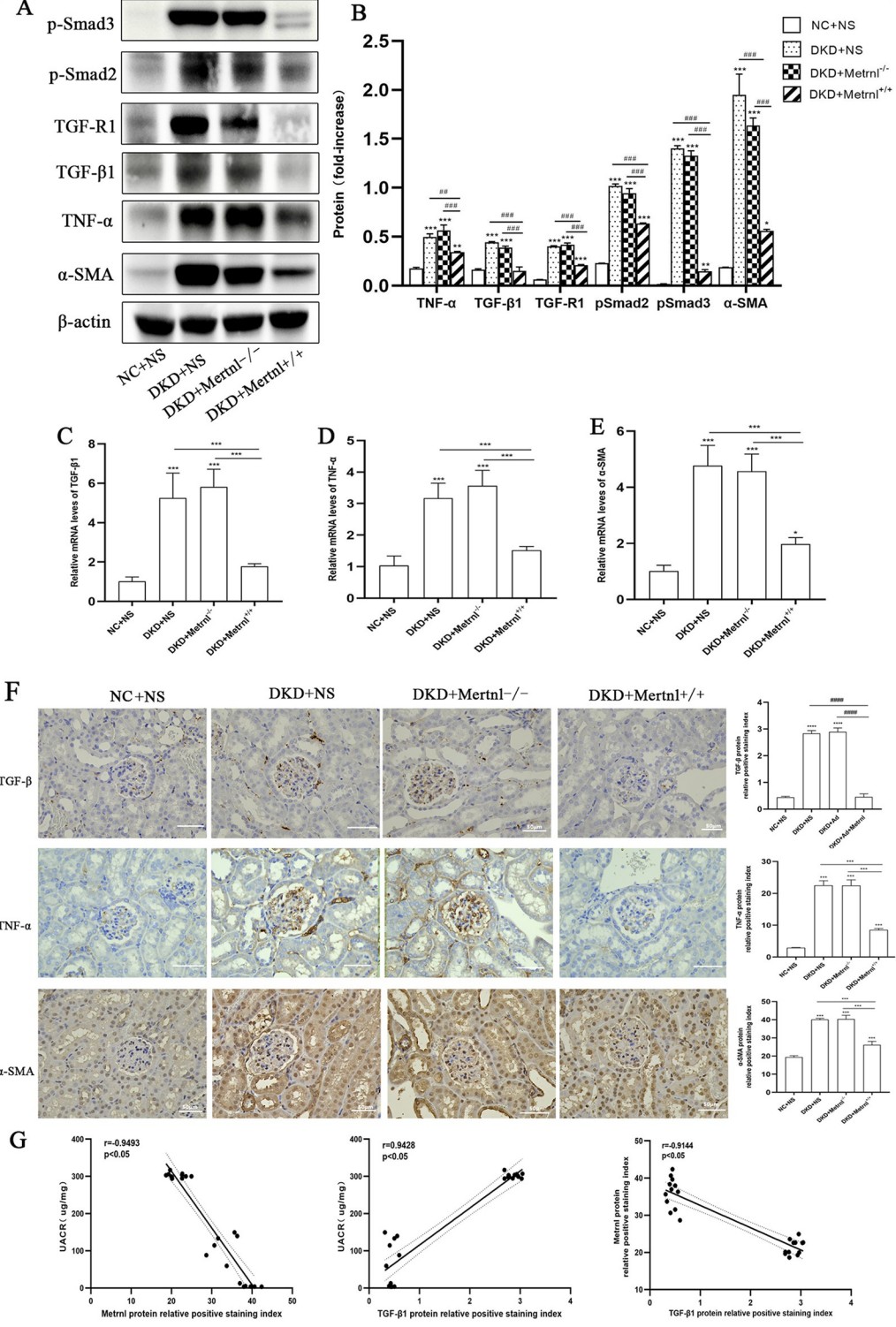

**Fig 6. The TGF-β1/Smads signaling pathway and α-SMA expression levels in mice.** (A, B) Western blot analysis was performed to detect the relative expression levels of TGF-β1, TNF-α, and α-SMA in the kidney tissues of mice in all groups. (C-E) qRT-PCR was used to detect the relative expression levels of TGF-β1, TNF-α, and α-SMA in the kidney tissues of mice from each group. (F) IHC analysis was performed to detect the protein expression levels of TGF-β1, TNF-α, and α-SMA in the kidney tissues of the mice from each group. Scale bar = 50μm. (G) Pearson's bivariate correlation analysis was conducted to assess correlations between the variables in the study. NC + NS group: normal control group,

DKD + NS group: diabetic nephropathy + saline group, DKD + Metrnl-/- group: diabetic nephropathy + control
adenovirus group, DKD + Metrnl$^{+/+}$ group: diabetic nephropathy + Metrnl overexpression adenovirus group. Compared
with the NC group, *P<0.05, ** P<0.01, ***P<0.001; Compared with DKD + Metrnl$^{+/+}$ group, #P<0.05, ##P<0.01,
###P<0.001, ####P<0.0001.

activation of proteins related to this pathway in the kidneys of DKD mice. Enhancing the
expression of Metrnl in the kidneys of DKD mice resulted in decreased levels of TGF-β1/Smad
pathway-related proteins, which subsequently alleviated renal damage.

DKD is a major cause of death and disability among patients with diabetes [20, 21]. In the
early stages, clinical manifestations of DKD may be subtle, and disease progression is slow
[22]. At this stage, the disease is both preventable and controllable [23]. However, if left
untreated, DKD can progress to ESKD, with approximately 20% of patients requiring dialysis
to sustain life [21, 24]. Unfortunately, there is currently a lack of targeted drugs to effectively
manage DKD progression. The db/db mouse model presents several advantages, including a
shorter incubation period, disease progression closely resembling the clinical manifestations of
DKD, and a lower mortality rate [25]. Therefore, we selected db/db mice as a model for this
study. Our findings indicate that FPG levels in the DKD group were consistently higher than
those in the normal control group. FPG levels gradually increased with age, stabilizing after 12
weeks. Similarly, the UACR in the morning urine of DKD mice progressively increased, with
microalbuminuria onset at 12 weeks and macroalbuminuria at 20 weeks. These findings sug-
gest that the db/db mouse model effectively mimics the development and progression of DKD,
making it suitable for studying the disease and evaluating therapeutic interventions.

The TGF-β1/Smads signaling pathway is a well-established pathway intimately associated
with normal physiological processes within the body [26]. Its abnormal activation has been
linked to the onset and progression of various fibrotic diseases [27]. In DKD, the TGF-β1/
Smads signaling pathway plays a pivotal role in renal fibrosis. Excessive activation of this path-
way leads to increased gene transcription of fibrotic molecules such as α-SMA and collagen
type I, exacerbating renal fibrotic lesions [9, 28]. Targeting the TGF-β1/Smads signaling path-
way has shown promise in regulating the progression of DKD. TGF-β blockers have been
reported to attenuate high glucose-induced podocyte injury and interfere with the TGF-β1/
Smads signaling pathway, thereby improving regulation of DKD progression [25, 26]. Consis-
tent with previous studies, our results demonstrated elevated expression levels of TNF-α,
TGF-β1, TGF-R1, pSmad2, pSmad3, and α-SMA in the kidney tissues of DKD mice compared
to those in the normal control group. Additionally, DKD mice exhibited enlarged glomerular
morphology, significant proliferation of mesangial cells, increased infiltration of inflammatory
cells in the tubulointerstitial space, thickening of the glomerular basement membrane, and
obvious proliferation of collagen fibers in the glomeruli and interstitium of the kidney. These
findings underscore the critical role of the TGF-β1/Smads signaling pathway in the develop-
ment of renal fibrotic lesions in DKD. However, the key factors involved in the regulation of
the TGF-β1/Smads signaling pathway in DKD have not been fully elucidated. Further research
is needed to identify these key factors and to explore their potential as therapeutic targets in
DKD.

Metrnl is a recently discovered secretory factor that has been extensively studied in the
endocrine metabolism [29]. It has been shown to play a role in repairing inflammatory dam-
age, improving the lipid profile, increasing insulin sensitivity, and reducing insulin resistance
[30, 31]. Clinical studies have found that reduced levels of Metrnl expression in patients
with DKD are negatively correlated with indicators of renal function deterioration [14]. Con-
sistent with these clinical findings, our animal experiments in this study also revealed the

downregulation of Metrnl expression in DKD mice compared to the normal control group. Moreover, when we injected an adenovirus carrying the Metrnl protein to overexpress Metrnl in DKD mice, we observed improvements in glomerular morphology, reduction in mesangial cell proliferation, decreased inflammatory cell infiltration in the kidney interstitium, reduced glomerular and interstitial collagen deposition, and decreased thickening of the renal basement membrane. Additionally, the number of foot process effacements was restored.

Further investigation revealed that overexpression of Metrnl inhibited the expression of TNF-α, TGF-β1, TGF-R1, pSmad2, pSmad3, and α-SMA. These findings suggest that Metrnl exerts a protective effect on DKD kidneys, potentially by inhibiting the expression of TGF-β1/Smads signaling pathway, thereby reducing the production of fibrotic molecules such as α-SMA. Interestingly, we found that in diabetic nephropathy, TGF-β was predominantly detected in the glomeruli, whereas α-SMA was primarily found in tubular lesions. TGF-β is a central mediator of the fibrotic response, and its activation in the glomeruli can initiate a cascade of pro-fibrotic signaling pathways that affect the entire renal architecture, including the tubules [32]. We suggest that the likely mechanism is that Metrnl-mediated suppression of TGF-β in glomeruli can prevent fibrosis in both glomeruli and tubules due to the interconnected nature of the kidney's structure and function.

This study has limitations that require acknowledgment. First, the sample size was relatively small, consisting of only 24 db/db mice and eight db/m mice in the control group. A larger sample size would provide greater statistical power and improve the generalizability of the findings. Second, while this study demonstrated that Metrnl ameliorated renal damage in DKD by inhibiting the TGF-β1/Smads signaling pathway, it is essential to note that we only assessed the expression of individual signaling molecules within the pathway. To gain a comprehensive understanding of the entire signaling pathway's role in DKD, it is necessary to intervene and manipulate the expression of each pathway component. Further research with comprehensive interventions targeting the TGF-β1/Smads signaling pathway would provide more definitive evidence of its involvement in DKD. These limitations highlight the need for further research with larger sample sizes and more extensive investigations into the mechanisms underlying the effects of Metrnl and the TGF-β1/Smads signaling pathway in DKD.

## 5. Conclusions

In conclusion, this study demonstrated the potential protective effect of Metrnl on DKD kidneys by inhibiting the TGF-β1/Smads signaling pathway and reducing the production of fibrotic molecules such as α-SMA. However, further investigation is required to ascertain whether Metrnl's renoprotective effects are primarily mediated through its action on the TGF-β1/Smads signaling pathway. Future studies should include knockdown experiments targeting critical proteins within this pathway to provide more definitive insights. Additionally, it is essential to explore whether Metrnl can delay the progression of DKD through alternative signaling pathways. Further research in these areas will provide a more comprehensive understanding of the mechanisms underlying the protective effects of Metrnl on DKD.

## Supporting information

**S1 Raw images. Original images for BlotsGels.**
(ZIP)

**S1 Data. The raw data.**
(RAR)

**S1 Graphical abstract.**
(PNG)

**S1 Checklist.** *PLOS ONE* **humane endpoints checklist.**
(DOCX)

## Author Contributions

**Conceptualization:** Lu Lin, Xin Lin.

**Formal analysis:** Lu Lin, Shulin Huang, Pin Chen.

**Funding acquisition:** Xiangjin Xu.

**Investigation:** Xin Lin, Pin Chen.

**Methodology:** Xin Lin, Xiaoling Liu, Chunmei Li, Pin Chen.

**Supervision:** Xiangjin Xu.

**Writing – original draft:** Lu Lin, Xin Lin, Chunmei Li.

**Writing – review & editing:** Shulin Huang, Xiaoling Liu, Xiangjin Xu, Pin Chen.

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
