## [Decision Letter · Decision Letter 0]

9 Jun 2024

PONE-D-24-14449Upregulation of Metrnl improves diabetic kidney disease by inhibiting the TGF-β1/Smads signaling pathway: A potential therapeutic targetPLOS ONE

Dear Dr. chen,

Thank you for submitting your manuscript to PLOS ONE. After careful consideration, we feel that it has merit but does not fully meet PLOS ONE’s publication criteria as it currently stands. Therefore, we invite you to submit a revised version of the manuscript that addresses the points raised during the review process.

Reviewer #1: Lin et al. investigated the role of Metrnl in diabetic kidney disease (DKD) using db/db mice and explored the possible mechanism involving the TGF-β1/Smad signaling pathway. They found that overexpression of Metrnl in db/db mice improved renal function and reduced the expression of inflammatory factors such as TNF-α, TGF-β1, and α-SMA. They also found that Pearson correlation analysis revealed a negative association between UACR and Metrnl, and a positive correlation between UACR and TGF-β1 in db/db mice. They concluded that upregulation of renal Metrnl expression may attenuate renal injury by modulating the TGF-β1/Smad signaling pathway in DKD. This study may shed light on potential therapeutic targets for the treatment of DKD in type 2 diabetes, but some concerns need to be addressed.

In 2.1, the authors should provide details of the adenoviral vectors used in this study. Are the adenoviral vectors designed to overexpress Metrnl in db/db mice systemically or in a tissue-specific manner driven by the promoters and enhancers?

In 2.4, the authors describe that the DKD+NS group received adenovirus dilutions in saline. The reviewer believes that only saline was injected in these control groups and this point should be verified. The authors also need to specify whether the NC group received the control adenovirus vector or saline.

In Figures 1C-D, the authors showed that overexpression of Metrnl in db/db mice resulted in a reduction of FBS and UACR. According to the previous paper cited as reference 13, BMI, HbA1c and ACR were reported to be associated with serum Metrnl in patients with type 2 diabetes mellitus. Why do you think that the db/db mice overexpressing Metrnl did not lose body weight as shown in Figure 1B? The reviewer also thinks that it would be better to show other biochemical parameters such as BUN, TG, HDL-C in db/db mice overexpressing Metrnl.

In Figures 3E and 4F, the authors evaluated changes in the expression of Metrnl, TGF-beta, TNF-alpha and alpha-SMA proteins expressed as a relative positive staining index. Authors should provide the procedures for their semi-quantitative analyses in the Materials and Methods section.

In Figure 4A-B, the thylakoid cells and inflammatory cells were difficult to see. It would be better to make the figures easier to understand, for example, by inserting arrows or asterisks.

The authors describe that glomerular size, thylakoid cell proliferation, and the number of inflammatory cells were reduced in db/db mice overexpressing Metrnl. The authors should provide semi-quantitative analysis data to support those descriptions.

The TEM images shown in Figures 4C-D are unclear and should be replaced.

In their interpretation of Figure 5, the authors use the term "significant" to describe pathological changes in db/db mice overexpressing Metrnl. However, the reviewers believe that the authors' descriptions are inappropriate because they have not provided semi-quantitative analysis data.

The results of TNF-alpha immunostaining shown in Figure 6F are unclear. They should be replaced.

In Figure S1, the authors describe that Mtrnl may attenuate the TGF-β/Smad signaling pathway in renal tissue and prevent fibrosis in glomeruli and tubules. However, in Figure 6, TGF-β protein was mainly detected in glomeruli and α-SMA protein was mainly detected in tubular lesions. What mechanisms do you think that Mtrnl-mediated suppression of TGF-β in glomeruli would prevent fibrosis in glomeruli and tubules?

In lines 443-444, the description should be corrected.

We look forward to receiving your revised manuscript.

Kind regards,

Md Shaifur Rahman, Ph.D

Academic Editor

PLOS ONE

Journal Requirements:

Reviewers' comments:

Reviewer's Responses to Questions

**Comments to the Author**

1. Is the manuscript technically sound, and do the data support the conclusions?

Reviewer #1: Partly

2. Has the statistical analysis been performed appropriately and rigorously? 

Reviewer #1: Yes

3. Have the authors made all data underlying the findings in their manuscript fully available?

Reviewer #1: No

4. Is the manuscript presented in an intelligible fashion and written in standard English?

Reviewer #1: Yes

5. Review Comments to the Author

Reviewer #1: Lin et al. investigated the role of Metrnl in diabetic kidney disease (DKD) using db/db mice and explored the possible mechanism involving the TGF-β1/Smad signaling pathway. They found that overexpression of Metrnl in db/db mice improved renal function and reduced the expression of inflammatory factors such as TNF-α, TGF-β1, and α-SMA. They also found that Pearson correlation analysis revealed a negative association between UACR and Metrnl, and a positive correlation between UACR and TGF-β1 in db/db mice. They concluded that upregulation of renal Metrnl expression may attenuate renal injury by modulating the TGF-β1/Smad signaling pathway in DKD. This study may shed light on potential therapeutic targets for the treatment of DKD in type 2 diabetes, but some concerns need to be addressed.

In 2.1, the authors should provide details of the adenoviral vectors used in this study. Are the adenoviral vectors designed to overexpress Metrnl in db/db mice systemically or in a tissue-specific manner driven by the promoters and enhancers?

In 2.4, the authors describe that the DKD+NS group received adenovirus dilutions in saline. The reviewer believes that only saline was injected in these control groups and this point should be verified. The authors also need to specify whether the NC group received the control adenovirus vector or saline.

In Figures 1C-D, the authors showed that overexpression of Metrnl in db/db mice resulted in a reduction of FBS and UACR. According to the previous paper cited as reference 13, BMI, HbA1c and ACR were reported to be associated with serum Metrnl in patients with type 2 diabetes mellitus. Why do you think that the db/db mice overexpressing Metrnl did not lose body weight as shown in Figure 1B? The reviewer also thinks that it would be better to show other biochemical parameters such as BUN, TG, HDL-C in db/db mice overexpressing Metrnl.

In Figures 3E and 4F, the authors evaluated changes in the expression of Metrnl, TGF-beta, TNF-alpha and alpha-SMA proteins expressed as a relative positive staining index. Authors should provide the procedures for their semi-quantitative analyses in the Materials and Methods section.

In Figure 4A-B, the thylakoid cells and inflammatory cells were difficult to see. It would be better to make the figures easier to understand, for example, by inserting arrows or asterisks.

The authors describe that glomerular size, thylakoid cell proliferation, and the number of inflammatory cells were reduced in db/db mice overexpressing Metrnl. The authors should provide semi-quantitative analysis data to support those descriptions.

The TEM images shown in Figures 4C-D are unclear and should be replaced.

In their interpretation of Figure 5, the authors use the term "significant" to describe pathological changes in db/db mice overexpressing Metrnl. However, the reviewers believe that the authors' descriptions are inappropriate because they have not provided semi-quantitative analysis data.

The results of TNF-alpha immunostaining shown in Figure 6F are unclear. They should be replaced.

In Figure S1, the authors describe that Mtrnl may attenuate the TGF-β/Smad signaling pathway in renal tissue and prevent fibrosis in glomeruli and tubules. However, in Figure 6, TGF-β protein was mainly detected in glomeruli and α-SMA protein was mainly detected in tubular lesions. What mechanisms do you think that Mtrnl-mediated suppression of TGF-β in glomeruli would prevent fibrosis in glomeruli and tubules?

In lines 443-444, the description should be corrected.

6. PLOS authors have the option to publish the peer review history of their article (what does this mean?). If published, this will include your full peer review and any attached files.

Reviewer #1: No

---

## [Author Response · Author response to Decision Letter 0]

18 Jun 2024

Editor-in-Chief

PLOS ONE

Dear Editor: 

We wish to re-submit our manuscript, titled “Upregulation of Metrnl improves diabetic kidney disease by inhibiting the TGF-β1/Smads signaling pathway: A potential therapeutic target” The manuscript ID is PONE-D-24-14449.

We thank you and the reviewers for your thoughtful suggestions and insights. The manuscript has benefited from these insightful suggestions. I look forward to working with you and the reviewers to move this manuscript closer to publication in PLOS ONE.

The manuscript has been rechecked, and the necessary changes have been made in accordance with the reviewers’ suggestions. The responses to all comments have been prepared and attached herewith. We have also ensured that the references are closely related to the content of this study.

Thank you for your consideration. I look forward to hearing from you.

Sincerely,

Pin Chen, 

Email address: chenpin@21cn.com

COMMENTS TO THE AUTHOR:

Reviewer #1: Lin et al. investigated the role of Metrnl in diabetic kidney disease (DKD) using db/db mice and explored the possible mechanism involving the TGF-β1/Smad signaling pathway. They found that overexpression of Metrnl in db/db mice improved renal function and reduced the expression of inflammatory factors such as TNF-α, TGF-β1, and α-SMA. They also found that Pearson correlation analysis revealed a negative association between UACR and Metrnl, and a positive correlation between UACR and TGF-β1 in db/db mice. They concluded that upregulation of renal Metrnl expression may attenuate renal injury by modulating the TGF-β1/Smad signaling pathway in DKD. This study may shed light on potential therapeutic targets for the treatment of DKD in type 2 diabetes, but some concerns need to be addressed.

1.In 2.1, the authors should provide details of the adenoviral vectors used in this study. Are the adenoviral vectors designed to overexpress Metrnl in db/db mice systemically or in a tissue-specific manner driven by the promoters and enhancers?

Response: Thank you for your comment. The adenoviral vectors are designed to overexpress Metrnl in db/db mice systemically. The corresponding details have been added to the manuscript and highlighted in red for your reference (marked in red, lines 79-81,89-102).

2.In 2.4, the authors describe that the DKD+NS group received adenovirus dilutions in saline. The reviewer believes that only saline was injected in these control groups and this point should be verified. The authors also need to specify whether the NC group received the control adenovirus vector or saline.

Response: Thank you for your comment. This is an error in expression. The correct wording is as follows: "Equal volumes of normal saline were given to the NC+NS and DKD+NS groups". The NC + NS group received normal saline. We have added this content to the manuscript to address this (marked in red, line131).

3.In Figures 1C-D, the authors showed that overexpression of Metrnl in db/db mice resulted in a reduction of FBS and UACR. According to the previous paper cited as reference 13, BMI, HbA1c and ACR were reported to be associated with serum Metrnl in patients with type 2 diabetes mellitus. Why do you think that the db/db mice overexpressing Metrnl did not lose body weight as shown in Figure 1B? The reviewer also thinks that it would be better to show other biochemical parameters such as BUN, TG, HDL-C in db/db mice overexpressing Metrnl.

Response: Thank you for your comment. In Figures 1C-D, we showed that overexpression of Metrnl in db/db mice resulted in a reduction of FBG and UACR. Although reference 13 reported associations between BMI, HbA1c, and ACR with serum Metrnl in patients with type 2 diabetes mellitus, our study did not observe a significant loss in body weight in db/db mice overexpressing Metrnl. This discrepancy could be due to species-specific metabolic responses or differences in experimental conditions. Additionally, we acknowledge the reviewer's suggestion and have now included data on BUN, TC, TG and HDL-C in the revised manuscript to provide a more comprehensive analysis (Figure 2E-H). 

4.In Figures 3E and 4F, the authors evaluated changes in the expression of Metrnl, TGF-beta, TNF-alpha and alpha-SMA proteins expressed as a relative positive staining index. Authors should provide the procedures for their semi-quantitative analyses in the Materials and Methods section.

Response: Thank you for your comment. In response to the reviewer's comment, we have added the procedures for the semi-quantitative analyses of the expression of Metrnl, TGF-beta, TNF-alpha, and alpha-SMA proteins, expressed as a relative positive staining index, to the Materials and Methods section. The details are now included in the revised manuscript (marked in red, lines 204-206).

5.In Figure 4A-B, the thylakoid cells and inflammatory cells were difficult to see. It would be better to make the figures easier to understand, for example, by inserting arrows or asterisks.

Response: Thank you for your comment. The details are now included in the revised manuscript and Figure 4A-B.

6.The authors describe that glomerular size, thylakoid cell proliferation, and the number of inflammatory cells were reduced in db/db mice overexpressing Metrnl. The authors should provide semi-quantitative analysis data to support those descriptions. The TEM images shown in Figures 4C-D are unclear and should be replaced.

Response: Thank you for your comment. We have provided the semi-quantitative analysis data. These details have been included in Figure 4, and we have also adjusted the image clarity for better visualization. (Figure 4E-F). (marked in red, lines 308-314).

7.In their interpretation of Figure 5, the authors use the term "significant" to describe pathological changes in db/db mice overexpressing Metrnl. However, the reviewers believe that the authors' descriptions are inappropriate because they have not provided semi-quantitative analysis data.

Response: Thank you for your comment. We have revised the manuscript to use more accurate terminology.

8.The results of TNF-alpha immunostaining shown in Figure 6F are unclear. They should be replaced.

Response: Thank you for your comment. We have adjusted the image clarity for better visualization. (Figure 6F). 

9.In Figure S1, the authors describe that Mtrnl may attenuate the TGF-β/Smad signaling pathway in renal tissue and prevent fibrosis in glomeruli and tubules. However, in Figure 6, TGF-β protein was mainly detected in glomeruli and α-SMA protein was mainly detected in tubular lesions. What mechanisms do you think that Mtrnl-mediated suppression of TGF-β in glomeruli would prevent fibrosis in glomeruli and tubules?

Response: Thank you for your insightful question. In Figure S1, we describe that Metrnl may attenuate the TGF-β/Smad signaling pathway in renal tissue and prevent fibrosis in glomeruli and tubules. In Figure 6, we observed that TGF-β was primarily detected in the glomeruli while α-SMA was mainly detected in tubular lesions.The Metrnl-mediated suppression of TGF-β in glomeruli can prevent fibrosis in both glomeruli and tubules due to the interconnected nature of the kidney's structure and function. TGF-β is a central mediator of the fibrotic response and its activation in the glomeruli can initiate a cascade of pro-fibrotic signaling pathways that affect the entire renal architecture, including the tubules. This crosstalk can lead to a broader anti-fibrotic effect. By inhibiting TGF-β signaling in the glomeruli, Metrnl may reduce the downstream pro-fibrotic signals, thereby attenuating overall renal fibrosis, affecting both glomerular and tubular compartments.These findings suggest that glomerular TGF-β suppression by Metrnl can have wider implications for renal health, preventing the progression of fibrosis through interconnected signaling pathways. We have added this content to the manuscript to address this (marked in red, lines 450-460).

10.In lines 443-444, the description should be corrected.

Response: In response to the reviewer's comment, we have corrected the description in lines 443-444 accordingly. Thank you for pointing this out.

END

---

## [Decision Letter · Decision Letter 1]

11 Jul 2024

PONE-D-24-14449R1Upregulation of Metrnl improves diabetic kidney disease by inhibiting the TGF-β1/Smads signaling pathway: A potential therapeutic targetPLOS ONE

Dear Dr. chen,

Thank you for submitting your manuscript to PLOS ONE. After careful consideration, we feel that it has merit but does not fully meet PLOS ONE’s publication criteria as it currently stands. Therefore, we invite you to submit a revised version of the manuscript that addresses the points raised during the review process.

1. The manuscript would need a partial review of the English language. The presenting content has minor spelling and grammatical errors, especially in the abstract.

2. Some of the sentences have no references, please revise and add their references. For example: "Several studies have examined the association between Metrnl level and adverse diabetic renal events in patients with DKD. These studies found a negative correlation between serum Metrnl concentration and the risk of DKD", Ref?

3. In the purpose section of the abstract, "This study aimed to investigate the role of Metrnl in this model" What do you mean by "this"?

4. It is suggested to add a box of abbreviations before the abstract, many of the abbreviations in the abstract are incomprehensible.

5. The mechanism of action of Metrnl in the control of diabetic kidneys is better depicted in a figure or graphical abstract.

6. Some explanations in the discussion are not needed. Please add the results of previously assessed studies and justify the obtained results. 

In addition to above, could you please clarify how you generated the Supp. fig. 2 (mouse with organ)

We look forward to receiving your revised manuscript.

Kind regards,

Md Shaifur Rahman, Ph.D

Academic Editor

PLOS ONE

Journal Requirements:

Additional Editor Comments:

Dear Authors,

Please consider the following points raised by reviewer.

1. The manuscript would need a partial review of the English language. The presenting content has minor spelling and grammatical errors, especially in the abstract.

2. Some of the sentences have no references, please revise and add their references. For example: "Several studies have examined the association between Metrnl level and adverse diabetic renal events in patients with DKD. These studies found a negative correlation between serum Metrnl concentration and the risk of DKD", Ref?

3. In the purpose section of the abstract, "This study aimed to investigate the role of Metrnl in this model" What do you mean by "this"?

4. It is suggested to add a box of abbreviations before the abstract, many of the abbreviations in the abstract are incomprehensible.

5. The mechanism of action of Metrnl in the control of diabetic kidneys is better depicted in a figure or graphical abstract.

6. Some explanations in the discussion are not needed. Please add the results of previously assessed studies and justify the obtained results.

In addition to above, could you please clarify how you generated the Supp. fig. 2 (mouse with organ)

Thanks.

Reviewers' comments:

Reviewer's Responses to Questions

**Comments to the Author**

1. If the authors have adequately addressed your comments raised in a previous round of review and you feel that this manuscript is now acceptable for publication, you may indicate that here to bypass the “Comments to the Author” section, enter your conflict of interest statement in the “Confidential to Editor” section, and submit your "Accept" recommendation.

Reviewer #2: All comments have been addressed

2. Is the manuscript technically sound, and do the data support the conclusions?

Reviewer #2: Yes

3. Has the statistical analysis been performed appropriately and rigorously? 

Reviewer #2: Yes

4. Have the authors made all data underlying the findings in their manuscript fully available?

Reviewer #2: Yes

5. Is the manuscript presented in an intelligible fashion and written in standard English?

Reviewer #2: No

6. Review Comments to the Author

Reviewer #2: 1. The manuscript would need a partial review of the English language. The presenting content has minor spelling and grammatical errors, especially in the abstract.

2. Some of the sentences have no references, please revise and add their references. For example: "Several studies have examined the association between Metrnl level and adverse diabetic renal events in patients with DKD. These studies found a negative correlation between serum Metrnl concentration and the risk of DKD", Ref?

3. In the purpose section of the abstract, "This study aimed to investigate the role of Metrnl in this model" What do you mean by "this"?

4. It is suggested to add a box of abbreviations before the abstract, many of the abbreviations in the abstract are incomprehensible.

5. The mechanism of action of Metrnl in the control of diabetic kidneys is better depicted in a figure or graphical abstract.

6. Some explanations in the discussion are not needed. Please add the results of previously assessed studies and justify the obtained results.

7. PLOS authors have the option to publish the peer review history of their article (what does this mean?). If published, this will include your full peer review and any attached files.

Reviewer #2: No

---

## [Author Response · Author response to Decision Letter 1]

14 Jul 2024

Dear Editor: 

We wish to re-submit our manuscript, titled “Upregulation of Metrnl improves diabetic kidney disease by inhibiting the TGF-β1/Smads signaling pathway: A potential therapeutic target” The manuscript ID is PONE-D-24-14449R1.

We thank you and the reviewers for your thoughtful suggestions and insights. The manuscript has benefited from these insightful suggestions. I look forward to working with you and the reviewers to move this manuscript closer to publication in PLOS ONE.

The manuscript has been rechecked, and the necessary changes have been made in accordance with the reviewers’ suggestions. The responses to all comments have been prepared and attached herewith. We have also ensured that the references are closely related to the content of this study.

Thank you for your consideration. I look forward to hearing from you.

Sincerely,

Pin Chen, 

Email address: chenpin@21cn.com

Reviewer : 

1.The manuscript would need a partial review of the English language. The presenting content has minor spelling and grammatical errors, especially in the abstract.

Response: Thank you for your valuable feedback. I have thoroughly reviewed and corrected the manuscript, particularly focusing on the spelling and grammatical errors you pointed out, especially in the abstract. The language has been refined to ensure clarity and accuracy throughout the manuscript.

2.Some of the sentences have no references, please revise and add their references. For example: "Several studies have examined the association between Metrnl level and adverse diabetic renal events in patients with DKD. These studies found a negative correlation between serum Metrnl concentration and the risk of DKD", Ref?

Response: Thank you for your insightful comments. I have revised the manuscript to include the necessary references for the sentences that previously lacked citations (marked in red, lines 41-42, 50-53, 63-65, 392-394, 406-407, 425-426. reference1, 8, 14, 22, 23, 26). The references have been carefully selected to support the claims made and ensure the accuracy and credibility of the manuscript.

3.In the purpose section of the abstract, "This study aimed to investigate the role of Metrnl in this model" What do you mean by "this"?

Response: Thank you for pointing out the need for clarification. In the purpose section of the abstract, "this model" refers to the specific experimental model used in the study to investigate the role of Metrnl. I have revised the sentence for better clarity: "This study aimed to investigate the role of Metrnl in an experimental model of diabetic kidney disease." (marked in red, lines 19-20)

4.It is suggested to add a box of abbreviations before the abstract, many of the abbreviations in the abstract are incomprehensible.

Response: Thank you for your helpful suggestion. I have added a box of abbreviations before the abstract to ensure that all abbreviations used are clearly explained and comprehensible to the readers (marked in red, lines 12-18). 

5.The mechanism of action of Metrnl in the control of diabetic kidneys is better depicted in a figure or graphical abstract.

Response: Thank you for your suggestion. I agree that a figure or graphical abstract would better depict the mechanism of action of Metrnl in the control of diabetic kidney disease. I have included a graphical abstract that visually represents the key mechanisms and findings of the study (Graphical abstract).

6.Some explanations in the discussion are not needed. Please add the results of previously assessed studies and justify the obtained results.

Response: Thank you for your valuable feedback. I have revised the discussion section to include the results of previously assessed studies. Additionally, I have provided a justification for the obtained results in the context of existing literature. This revision aims to enhance the discussion by linking our findings to the broader body of research and providing a comprehensive understanding of the study's implications.

7.In addition to above, could you please clarify how you generated the Supp. fig. 2 (mouse with organ)

Response: Thank you for your question regarding the generation of Supplementary Figure 2. The figure of the mouse with organs was created using the BioRender website. The entire image is original and designed specifically for this study.

END

---

## [Decision Letter · Decision Letter 2]

9 Aug 2024

Upregulation of Metrnl improves diabetic kidney disease by inhibiting the TGF-β1/Smads signaling pathway: A potential therapeutic target

PONE-D-24-14449R2

Dear Dr. Chen,

We’re pleased to inform you that your manuscript has been judged scientifically suitable for publication and will be formally accepted for publication once it meets all outstanding technical requirements.

Kind regards,

Md Shaifur Rahman, Ph.D

Academic Editor

PLOS ONE

Additional Editor Comments (optional):

Reviewers' comments:

Reviewer's Responses to Questions

**Comments to the Author**

1. If the authors have adequately addressed your comments raised in a previous round of review and you feel that this manuscript is now acceptable for publication, you may indicate that here to bypass the “Comments to the Author” section, enter your conflict of interest statement in the “Confidential to Editor” section, and submit your "Accept" recommendation.

Reviewer #3: All comments have been addressed

2. Is the manuscript technically sound, and do the data support the conclusions?

Reviewer #3: Partly

3. Has the statistical analysis been performed appropriately and rigorously? 

Reviewer #3: Yes

4. Have the authors made all data underlying the findings in their manuscript fully available?

Reviewer #3: No

5. Is the manuscript presented in an intelligible fashion and written in standard English?

Reviewer #3: Yes

6. Review Comments to the Author

Reviewer #3: Review Comments to the Author:

Title:

Upregulation of Metrnl improves diabetic kidney disease by inhibiting the TGF-β1/Smads signaling pathway: A potential therapeutic target

There are no new comments for authors

Thanks

7. PLOS authors have the option to publish the peer review history of their article (what does this mean?). If published, this will include your full peer review and any attached files.

Reviewer #3: No

---

## [Editor Report · Acceptance letter]

15 Aug 2024

PONE-D-24-14449R2 

PLOS ONE

Dear Dr. Chen, 

I'm pleased to inform you that your manuscript has been deemed suitable for publication in PLOS ONE. Congratulations! Your manuscript is now being handed over to our production team.

Kind regards, 

on behalf of

Dr. Md Shaifur Rahman 

Academic Editor

PLOS ONE